# Access to Public Support Systems Related to Severity of Adversities and Resilience among Adolescents

**DOI:** 10.3390/children9070948

**Published:** 2022-06-25

**Authors:** Lihong Huang, Svein Mossige, May-Britt Solem

**Affiliations:** 1Norwegian Social Research (NOVA), Oslo Metropolitan University, 0170 Oslo, Norway; 2Department of Psychology, University of Oslo, 0373 Oslo, Norway; sveinmos@psykologi.uio.no; 3Department of Social Work, Child Welfare and Social Policy, Oslo Metropolitan University, 0170 Oslo, Norway; maybr@oslomet.no

**Keywords:** child welfare services, mental health services for children and youth, pedagogical psychology services, resilience, self-harm, victimization of violence

## Abstract

Access to support systems is crucial for providing immediate assistance and treatment to children to counteract the long-term detrimental effects of various forms of violence. This study examines how adversity such as victimization of violence and self-injury behaviors among young people with their individual resilience is related to their access to support systems. The data used in our analysis are from two national youth surveys carried out in Norway in 2007 and 2015. We ask: To what degree do young people with experiences of violence gain access to support systems such as child welfare services, mental health services for children and youth, and pedagogical psychology services? Our results show that although not all young people who need help have gained access to support systems, victimization of violence and self-injury behavior significantly increase the likelihood of accessing these support systems. Our results also reveal a persistent effect of young people’s home socio-economic background on their unequal access to system support. More future research is needed on the subtle mechanisms and social–emotional implications of individual accessing system support from the macro-societal level and meso-system/family level.

## 1. Introduction

The prevalence of children in high-income countries being exposed to adversities and the consequences of these experiences may indicate the proportion of children needing psychological and social support and interventions [1]. Recent studies in Norway on the prevalence of children and young people reporting various forms of violence and abuse, including verbal abuse, physical abuse, sexual abuse, and peer bullying, provide a worrying depiction [2,3]. Between the ages of 12 and 16 years old, every fifth boy and girl experience or witness domestic violence by parents or between parents. About 5% of them experience severe physical violence, and over 6% are sexually abused by an adult [4]. By the age of 18–19, over half of all Norwegian youths will have experienced some form of abuse by peers or at home, and 20% will have been victims of multiple forms of abuse [2].

Research evidence shows that victimization of violence and abuse during childhood and adolescence has severe detrimental effects on the mental health of the victims such as anxiety, depression, self-harm, and impaired mental health [5,6]. Self-harm may be understood as a form of violence directed towards oneself and as a coping mechanism, often related to other challenges [7]. It is associated with many kinds of serious negative experiences such as domestic violence, sexual abuse, and a dysfunctional family environment [5,8], and is an indicator of mental burden and psychological stress [9]. The majority of individuals who engage in self-harm rarely present to clinical services specifically for their self-harm [10]. Many of them either do not seek help or drop out of treatment [8,11].

The prevalence of mental health problems among children and adolescents is around 14% [3,12,13], of which behavioral and emotional problems characterized by anxiety and depression are the largest group. In addition, a recent regional youth survey conducted in Norway showed that a quarter of adolescent boys and half of girls feel a lot of stress at school due to academic achievement, while pressure also appears to be connected to demands or expectations of being good at sports (16% boys and 25% girls) and receiving likes on social media (4% boys and 15% girls). One in ten Norwegian adolescents report having difficulties handling these issues [12]. The reported prevalence of mental health problems makes it relevant to ask whether there is a mismatch between the number of young people struggling with mental health problems and those referred to mental health services for children to receive professional help.

Access to support systems is crucial for providing immediate assistance and treatment to children in need. To gain such access may also be an expression of resilience and contribute to building resilience. Resilience is understood as a personality trait inherent to an individual and as a process or phenomenon influenced by culture and context [13,14,15]. The concept of resilience is particularly relevant for children and adolescents who are victims of violence, as resilience may indicate an individual’s capacity to navigate their way towards health-supporting resources. Meanwhile, resilience also depends on an individual’s access to resources in their family, community, and culture, which provide health resources and experiences in the face of adversity.

Resilience highlights that access to support systems may provide the resources that individuals need to cope, rather than the capacities of individuals themselves. The term implies how society facilitates access to support systems for young people. Previous research evidence shows that the most effective interventions for promoting resilience are those that address multiple systems [13,14,15]. Resilience may affect young people’s mental health in different ways. First, if exposed to adversities, resilience may imply that individuals do not show the same degree of mental health problems that would be expected otherwise. Viewing resilience from this perspective, there must be exposure to serious adversities and more moderate measures of effects than expected. Second, if individuals are exposed to adversities that lead to mental health problems, resilience could appear through them having access to support systems for children and youth. A third possible function is that those who are resilient are less inclined to being exposed to serious adversities. This function is difficult to measure, however. In this study, our concern is whether resilience appears to matter for access to support systems among those reporting mental health problems and having been exposed to adversities.

Previous research evidence shows that the most effective interventions for promoting resilience are those that address multiple systems. Systemic influences matter as much as individual factors, at least in terms of positive outcomes [13,14]. There is increasing evidence showing that interpersonal and relational, protective, and promotive factors, as well as those that function at formal and institutional levels, promote youth resilience. A multisystem, socioecological model of resilience emphasizes the social systems that facilitate access to the resources that individuals need to cope, rather than the capacities of individuals themselves. An unsatisfactory protection service significantly limits the capacity to foster resilience among young victims of violence and abuse without a solid family and social support system to begin with [16]. This linkage between macro-system level, meso-institution/family level and micro-individual level resilience can be explained using the concept of social resilience [17]. Previous research has demonstrated that multisystem resilience that is context-dependent needs to be considered when supporting young people living with adversity [18]. In addition, previous studies have found that family resilience is a significant mediator of local community resources for child wellbeing [19,20].

Using data from two national youth surveys in Norway on the victimization of violence and abuse from 2007 [21] and 2015 [22], this study examines the resilience of young people living with adversity such as victimization of violence and self-injury behaviors and their access to support systems such as child welfare services, mental health services for children and youth, and pedagogical psychology services. First, we present the three social support systems that may be involved in providing support to young people living with adversities. Then, we introduce the data and methods of analysis, before presenting the results. After a discussion of the results, we provide the conclusions and implications of the study.

### The Norwegian Support Systems for Children and Youth Living with Adversity

Traditionally working at separate levels of administration, three different types of support systems focus on helping children and youth living with adversity: child welfare services, mental health services for children and youth, and pedagogical psychology services. Professionals in frontline services have the important task of flagging up vulnerable children and have the possibility of preventing adversity, giving the child help, or referring them to a specialist service. It has become public knowledge that thousands of children who are victims of violence, abuse, and neglect do not receive timely or adequate protection and support [23]. There has been strong advocacy from the state, researchers, specialists, and local community practitioners on close professional cooperation and collaboration between the three agencies, which resulted in numerous local initiatives of interprofessional collaboration when working with children and youths during the 2010s [24,25,26]. At present, interprofessional collaboration in the Norwegian welfare service is still an ongoing effort that encounters difficulties due to previously established individual service autonomy and segregation.

Upon being notified by a neighbor, a teacher, a child him/herself, or a family doctor about children aged 0–18 years old who may have a difficult life situation, the child welfare services at the municipality level have the duty to investigate this situation within 3 months and recommend how to help the child and the family in the best way possible for the child. Nearly 3% of Norwegian children receive interventions from child welfare services each year (Statistics Norway 2020, available online: https://www.ssb.no/sosiale-forhold-og-kriminalitet/artikler-og-publikasjoner/4-av-5-i-det-kommunale-barnevernet-har-hoyere-utdanning, accessed on 20 April 2022). During the last century, child welfare services worked mostly on removing children from problematic homes where parents were drug addicts, violent and abusive, poor, or had low educational attainment. In recent years, some of these practices have been placed under serious criticism as being severely socially and racially biased; socioeconomic disadvantages and adversities follow both children in care and after care into adulthood [27,28]. Hence, there has been a public outcry for child welfare services to change from punitive to supportive. In the last 20 years, child welfare services have undergone a series of systematic changes to strengthen professional competence and become “research- and evidence-based organs of the welfare system”, with assessments and decisions based on rationality and accountability associated with science [29]. Most employees working in child welfare services (80%) have bachelor’s or master’s degrees, and 65% have a professional training background, mostly in social work, child protection pedagogy, psychology, or other academic training. In the same process of professionalism, child welfare services have undergone a dramatic shift in treatment of cases, among which a minority (18%) involved removal of the child from the home and a majority (over 80%) provided in-home assistance and initiatives to help the child and the parents (Statistics Norway 2020, available online: https://www.ssb.no/sosiale-forhold-og-kriminalitet/artikler-og-publikasjoner/4-av-5-i-det-kommunale-barnevernet-har-hoyere-utdanning, accessed on 20 April 2022).

Mental health services for children and youth (BUP) are organized into interdisciplinary treatment teams consisting of doctors, psychologists, educators, social workers, and milieu personnel to treat children aged 0–18. The team is expected to provide a broad-spectrum approach, including the whole family as a target for the interventions. Unfortunately, services have been oriented more towards individual interventions, and a systemic approach has been difficult to apply due to a lack of available resources and professionals [3,30]. Services are characterized by interdisciplinarity, which provides a broad-spectrum approach, including the whole family in understanding and treating the problems. Usually, children are required to display certain symptoms to fulfill the criteria for a specific diagnosis to receive help from these services.

The pedagogical psychology service is the municipality’s advisory and expert body regarding children and young people’s needs for special pedagogical help in kindergarten and schools. Professionals working in this service often have several different areas of expertise (e.g., social work, psychology, law), which separate them from other social workers in the field of child protection [31]. Operating under the regulations of the Education Act, this service investigates the needs of children and provides advice and guidance to the school, kindergarten, and parents. Reasons for referrals often concern children’s emotional problems that impede learning. As part of the school’s support system, PPT has become increasingly focused on educational problems, while emphasis on children’s mental health has weakened (Utdanningsdirektoratet, available online: https://www.udir.no/kvalitet-og-kompetanse/samarbeid/pp-tjenesten/hva-gjor-pp-tjenesten, accessed on 24 April 2022).

## 2. Data and Methods

Data were obtained from two rounds of the Norwegian national youth survey on violence and abuse from 2007 and 2015 [2]. Both surveys used a stratified national sample of upper second schools, of which all students in their final year of education, aged 18 years and older, were invited to participate in the surveys. Both surveys followed strictly the Norwegian research ethic regulation approved by the Norwegian Data Protection Service (NSD) (Ref. 14/01407-5/EOL) using informed consent prior to the survey and anonymous procedure for insuring data safety and reusability. Data are available at Voldprogrammet—Forskningsprogram om vold i nære relasjoner (in English: The Domestic Violence Research Program—A research program about violence in close relationships; https://uni.oslomet.no/voldsprogrammet/ (accessed on 20 April 2022)). These survey data were stripped of several background variables before being made fully accessible without restriction. Therefore, as per our usual practice, data are safely stored and accessible from the NOVA—OsloMet Institutional Data Access/Ethics Committee, which grants researchers access if they meet the criteria to access confidential data.

### 2.1. Participants

Overall, 7033 students responded to the 2007 survey, with a response rate of 77.3%; conversely, 4531 students responded to the 2015 survey, which provided a sufficient response rate of 66.2%. Previous analyses [32] comparing the two samples found no systematic nor significant differences in the proportion of girls, the proportion of students from immigrant backgrounds, or the proportion of participants who reported that both parents were unemployed. In both datasets, nearly all respondents (99.8% in 2007 and 100% in 2015) were 18 years of age or older, and 58% of respondents were female.

### 2.2. Variables of Interest

Access to Support Systems: The surveys in both 2007 and 2015 included a question asking the young people “Have you ever been in contact with the following?” with “yes” or “no” responses on the three types of services providing support and help for children and youth living with adversity: child welfare services (Barnevernet), mental health services for children and youth (BUP), and the pedagogical psychology service (PPT). Unfortunately, as both surveys were cross-sectional and there were no follow-up questions on any details about contact with the support systems, we can only use these contacts to indicate young people’s access to them.

Measure of Resilience: As a measure of resilience, we incorporated the 28-item READ scale used in both surveys. Responses range from 1 (completely disagree) to 5 (completely agree) on a five-point Likert scale. The READ scale has five dimensions of resilience, which were tested among Norwegian youth: “family cohesion”, “personal competence”, “social competence”, “social resources”, and “structured style” [33]. Overall, the READ scale had the same mean between the two surveys; however, from 2007 to 2015, we observed significant changes in means in terms of an increase in dimensions of structured style and family cohesion and a decrease in dimensions of social competence and social resources [34].

Mental Health Problems: These were measured by a 12-item short version of the Hopkins Symptom Checklist (HSCL) [35], which indicated whether various symptoms of depression and anxiety were experienced in the past week. The item responses fell on a four-point scale, with responses ranging from “1” (“not been troubled at all”), to “2” (“been a little troubled”), “3” (“been quite troubled”), and “4” (“been very much troubled”). The psychological problems variable was indicated by the mean score from the sum of the 12 HSCL items. Higher values indicate poorer psychological health (or higher levels of psychological problems).

Measures of Victimization: Both the 2007 and 2015 surveys used similar questions to obtained data on all types of offenses against children, from which we constructed a variable including all forms of victimization with five categories: 0 = never have had any victimization of violence, 1 = victimization of a single form of violence, 2 = victimization of two forms of violence, 3 = victimization of three forms of violence, and 4 = victimization of four forms of violence [34].

Measures of Self-Harm: We included two categories of self-harm behavior: intentionally injuring oneself without intention to die (NSSI) and intentionally injuring oneself intending to die (SSI). NSSI is measured by counting responses of “yes, once” and “yes, more than once” to at least one of three questions: (1) “Have you at any time intentionally taken an overdose of pills or other medicine?”, (2) “Have you at any time tried to hurt yourself, e.g., cut yourself?” and (3) “Have you at any time ended up in the hospital due to an injury you have done to yourself intentionally?”. SSI is measured by counting responses of “yes, once” and “yes, more than once” to at least one of two questions: (1) “Have you at any time tried to kill yourself?” and (2) “Have you at any time ended up in hospital because you tried to kill yourself?”. We constructed a variable combining NSSI with SSI with three categories: 0 = never done self-injury, 1 = have done one of the two forms of self-injury, and 2 = have done both forms of self-injury.

Background Variables: Besides gender, we also looked at several home background factors. “Parents live together” is measured by parents’ civil status, combining the categories of “married” and “cohabitating” as “1”, or as “0” if not falling under these two categories. “Both parents with higher education” is measured by educational attainment at tertiary level by both parents as “1” or as “0”. Home finance is a subjective measure determined by asking the correspondent, “Has your family been in a good or bad financial situation in the past two years?” on a six-point scale (from 1 = a lot of ups and downs, 2 = financial difficulties all the time, 3 = financial difficulties most of the time, 4 = neither good nor bad, 5 = financially comfortable most of the time, 6 = financially comfortable all the time). “Family usually has good financial situation” is coded as “1” for responses “5” and “6” or as “0” for responses “1” to “4”. In both surveys, the respondent was asked to place the birth country of their father and mother among seven choices: Norway, another Nordic country, another European country, Asia, Africa, South America, and North America/Oceania. “Both parents are immigrants” is a combination of father’s and mother’s birthplace in countries other than Norway and any other Nordic country.

Table 1 presents descriptions of all the variables included in our analysis, with significant differences between the two survey time points marked.

### 2.3. Analysis Plan

We present our analysis in three steps, applying the concept of social resilience, which refers to a successful society that has the “institutional and cultural resources that groups and individuals mobilize to sustain their well-being” [17]. We look into the associations between individual factors, family resources and access to systemic resources available in the Norwegian children and youth welfare context. First, we present the descriptive results in Table 1 alongside a discussion of prevalence and trends/changes in issues of interest: the victimization of violence, self-harm, mental health problems, access to support systems, and five dimensions of resilience among Norwegian youths. Second, we use logistic regression analysis to determine which variables of background, victimization, and self-harm behavior increase or reduce the chances of young people’s access to each of the support systems. Third, we present the descriptive statistics of all factors, including five resilience dimensions, comparing young people who have had access to any support system with those who have never accessed them. In addition to the background variables and variables of severe adversity, we include the means of mental health problems and five resilience dimensions in a logistic regression model to detect the significant factors affecting young people’s access to support systems. Due to the significant changes observed in several variables between the two surveys (shown in Table 1), we decide to separate our analyses into 2007 and 2015 data and compare the effects of variables between the two time points as an effort to acknowledge the changes in the macro context in Norway, as described above.

## 3. Results

### 3.1. Prevalence and Trends/Changes over Time

As shown in Table 1, eight years apart, there are small but significant differences in distributions of most of the variables between the two cohorts of Norwegian youths. First, 3.6% more young people reported a better financial situation at home in 2015 than in 2007, while slightly fewer (2%) of their parents lived together. Second, the prevalence of victimization in 2015 was significantly lower for all forms of violence except verbal abuse than in 2007. We previously notice a general reduction in all forms of violence in young people’s homes and beyond, except for an increase in verbal abuse by peers (see [34]). Furthermore, as previous analysis [34] has identified, many young people are victims of multiple forms of violence. The prevalence of victimization of one or two forms of violence has not changed from 2007 to 2015, while the prevalence of victimization of three or four forms of violence has slightly reduced between the eight years.

Third, following a pattern similar to the one shown in Table 1, we notice a decrease (by 2.8%) in the proportions of young people who reported intentional self-injury or suicide attempt (by 4.4%) between 2007 and 2015. The proportion of young people reporting one of two forms of self-harm behaviors is at the same level in 2007 and 2015. However, there is a decrease in those who have engaged in both forms of self-harm, from 4.9% in the 2007 sample to 1.5% in the 2015 sample. In accordance with this, we also observe a significant and clear decrease in the proportion of young people who injured themselves “with the intent to die” from 6.0% in 2007 to 1.6% in 2015. The mean resilience score of all 28 items from the READ scale was 3.95 in both surveys (SD = 0.59 in the 2007 survey and SD = 0.66 in the 2015 survey), but all five dimensions of resilience have shown small but significant changes from 2007 to 2015. Family cohesion and the structured style of young people increased, while young people’s resilience in terms of personal competence, social competence and social resources decreased.

Between the eight years, we see a slight decrease in reported adversity experiences of young people and a small increase in their access to support systems (Table 1). When we count the support systems in Table 2, in total, around one-fifth of both samples have been in contact with them (18.3% in 2007 and 20.7% in 2015). While the access rate of only one type (Pedagogical Psychology Service) of the three supporting services remains the same between the eight years, there is a significant increase in the number of young people accessing two and three types of service. The proportion of young people who use child welfare services has increased by 1.8% and, for mental health services for children and youth, by 5.3%. There is a small but significant increase in reported level of mental health problems from 2007 to 2015.

### 3.2. Likelihood of Young People’s Access to System Support

Table 3 presents logistic regression analyses of young people’s background variables, violence victimization, and self-injury behaviors in relation to their access to support systems (child welfare services, mental health services for children and youth, and pedagogical psychology services). Among the background variables, being a female increases the likelihood of encountering children and youth mental health services by 57% in 2007 and by 30% in 2015. In the 2015 survey, having parents with higher education reduces the likelihood of contact with child welfare services by 41%.

Young people living in homes with a good financial situation and those whose parents live together have a significantly lower likelihood of contact with all three types of support systems, particularly child welfare services. Having immigrant parents significantly increases the odds of contact with child welfare services: by 97% in 2007 and 78% in 2015; however, it reduces the likelihood of contact with mental health services for children and youth, and has no effect on the likelihood of young people accessing pedagogical psychology services.

As expected, the odds ratios of accessing support systems doubled and tripled along with increases in victimization of violence. In particular, the odds ratio of access to child welfare services increases around three times for victims of two forms of violence, by four to six times for those victimized by three forms of violence, and by around eight times for victims of four forms of violence. For young people with any self-injury behavior, the odds ratio of contacting support systems also doubled. Meanwhile, for young people with both self-injury and suicide attempt history, from 2007 to 2015, there is a considerable increase in odds ratios of their access to child welfare services (2.58 in 2007; 5.46 in 2015) and mental health services for children and youth (9.79 in 2007; 16.14 in 2015). Similarly, we notice significant increases in odds ratios during the same years for victims of three or four forms of violence in relation to their access to child welfare services and mental health services.

In line with previous studies, we find that young people with parents of higher education, parents who live together, and parents with good finances decrease the likelihood of contact with the support systems. One plausible explanation could be that all these three background variables have a protective or resilient effect by lowering the need for young people from these families to be in contact with any of the support systems.

### 3.3. Resilience in Relation to Access to Support Systems

Table 4 presents descriptions of the background factors, violence victimization, self-harm behaviors, mental health problems and resilience in five dimensions, comparing those who have never had access to any and those have had access to one or more of the support systems (18.3% in the 2007 sample and 20.7% in the 2015 sample). The descriptive results show a similar pattern in both cohorts and align with what we observe in Table 3. Those who have had access to any support system, in comparison with those who have never had access to any support system, appear to over-represent disadvantaged families, where over half of the parents are separated and the family tends to have a poor financial situation. Victims of multiple forms of violence and young people with severe self-harm behaviors have significantly higher representation among those who have accessed support systems before compared to those who have never had access to them. Compared with those who have never accessed them, young people who have accessed these support systems previously have a significantly higher mean score for mental health problems and significantly lower mean scores in all five dimensions of resilience.

As several variables in Table 4 have significant correlations with each other, e.g., the five resilience dimensions, we used a stepwise method to solve problems of collinearity in the logistic regression analysis. Table 4 shows only the statistically significant associations of the final model on young people’s access to the support systems. Among the six variables with a significant effect in both 2007 and 2015, parents living together and family with a good financial situation have the same moderate negative effect on accessing support systems. Severe self-harm has the strongest positive odds in both surveys, and there is a significant increase in odds from 2007 to 2015 (ExpB = 4.38 with 95% CI: 3.28, 5.84 in 2007 and ExpB = 11.28 with 95% CI: 5.52, 23.07 in 2015). However, poor mental health indicates a significant likelihood of accessing support systems, but the odds decreased from 2007 to 2015. Of all five dimensions of youth resilience, only the dimension of family cohesion decreases the odds of accessing support systems, which is in line with previous findings [19] and is similar to the effects of advantaged home background variables in this study.

## 4. Discussion

Our analysis finds that the exposure to serious adversities strongly increases the likelihood of access to support systems. This indicates that, in reality, severe self-injury is among the strongest indications to be picked up by support systems, although the likelihood of young people with polyvictimization accessing them is also high and increased from 2007 to 2015. This can be observed in the context of a society becoming more open and aware of young people struggling with mental health issues and the existing fact that many young people experience multiple forms of violence. It can be viewed as a positive trend that there is an increase in access to support systems, while the proportion of young people living with severe adversities in Norway is decreasing. Both conditions may contribute towards creating social resilience [36] for society, support systems and the young people living with severe adversity. Social resilience implies that both adults and children need to be active, aware of, understand, and use available resources in their social context that are relevant to combating mental health problems among young people and violence against children.

However, despite the small positive trend from 2007 to 2015, our analysis reveals three worrying aspects of unsuccessful system support for young people and children living with adversity in Norway: First, there is still a substantial proportion of young people who are victims of violence and abuse. Second, not all young people living with violence and self-harm have access to support systems. Third, access to support systems is still subtly “steered” by socioeconomic advantages. Moreover, due to data limitations, we cannot assume that access to support systems has helped with building resilience in young people and their families, as part of creating social resilience. Additionally, due to data limitations, our study is not able explain the alarming albeit small decreases in youth resilience dimensions of personal and social competences and social resources in Norwegian society from 2007 to 2015.

What we discovered in this study is not unique—in fact, the lessons learned here may be related to studies from elsewhere. For example, family resilience can have a direct effect on child health and an indirect effect through mediating neighborhood resources [19]; multiple, historic, and interconnected support systems may also perpetuate adversity across time through traumatic impacts on both children and parents [37]. Many people experience adversities during their lifetime, even in a highly developed country such as Norway with advanced democratic and social welfare systems. Some need assistance from systemic services. Some recover from the problem on their own or with the support of family and friends. Some pass on the problems and issues to the next generation [37]. It is essential for the resilience of a society to provide timely multisystemic support to young people and children living with adversity. However, it is not a question of having general resistance resources available, but the capacity to actively use the resources one has (e.g., mental health services for children and youth, pedagogical psychology services or child welfare services). An individual’s experience of connection, based on cognitive, behavioral, and motivational factors [38], leads to the capacity to use their resistance resources.

## 5. Conclusions

Overall, from 2007 to 2015, our analyses show a decrease in the prevalence of some forms of victimization of violence and a decrease in the number of young people as victims of multiple forms of violence. A decrease in self-harm prevalence is also demonstrated. At the same time, there is a small but significant increase in the proportion of young people with access to support systems such as child welfare services and mental health services for children and youth. An advantaged home background—indicated by parents having higher levels of education, healthy financial situation in the family, and parents living together—continue to reduce the likelihood of accessing any system support. In 2007 and 2015, migrant status increased the likelihood of being in contact with child welfare services, while the odds of accessing mental health services reduced. However, in both surveys, the likelihood of accessing all types of support systems doubled and tripled for all young people that experienced victimization of violence and self-harm. The odds even increased by up to 7 to 8 times in access to child welfare services for young victims of 3–4 forms of violence, and by up to 16 times in access to mental health service for young people who engaged in both forms of self-injury. However, our analysis reveals a continuous service provision gap between young people living with adversity and those who have had access to these systems as well as a persistent effect of socio-economic background showing social inequalities of access.

In order to build social resilience, greater openness, transparent practice and close collaboration are necessary in policy, education and practice for all levels involved, from the government, support system operators, schools and families to individual children and young people. Future research should focus on the interlinkages between factors of the macro-, meso- and micro-levels, particularly both on the mechanisms and social–emotional implications of young people’s access to support systems, and on the process and quality of system support for those who have gained access.

## Figures and Tables

**Table 1 children-09-00948-t001:** Descriptions of variables used in this study.

Categories	Variables	2007 (N = 7033)	2015 (N = 4531)
Individual background	Female	58.1%	58.4%
	Both parents have higher education	36.1%	35.3%
	Parents live together *	66.3%	64.3%
	Family usually has good financial situation *	70.8%	74.4%
	Both parents are immigrants	8.8%	9.3%
Victimization of violence ^#^	Have ever been victims of verbal abuse *	37.4%	44.0%
	Have ever been witness to domestic violence *	39.1%	29.8%
	Have ever been a victim of physical abuse *	40.6%	32.3%
	Have ever been a victim of sexual abuse *	21.3%	19.5%
	Never have had any victimization of violence	33.2%	36.9%
	Victimization of a single form of violence	24.6%	24.9%
	Victimization of two forms of violence	19.7%	19.2%
	Victimization of three forms of violence *	15.6%	13.5%
	Victimization of four forms of violence *	6.9%	5.5%
Self-harm	Have ever intentionally injured self by cutting or overdose *	18.1%	15.3%
	Have ever injured self with the intent to die *	6.0%	1.6%
	Never done self-injury *	80.8%	84.6%
	Have done one of the two forms of self-injury	14.3%	13.9%
	Have done both forms of self-injury *	4.9%	1.5%
Support systems	Have ever been in contact with child welfare services *	5.3%	7.0%
	Have ever been in contact with mental health services for children and youth *	7.1%	12.4%
	Have ever been in contact with a pedagogical psychology service	11.2%	10.4%
Mental health problems	Mean score of Hopkins Symptom Checklist (HSCL) (Standard Deviation, SD) *	1.60 (0.56)	1.66 (0.56)
Resilience	Mean score of all 28 items from the READ scale (SD)	3.95 (0.59)	3.95 (0.66)
	Mean score of 6 items: family cohesion (SD) *	4.06 (0.79)	4.12 (0.82)
	Mean score of 8 items: personal competence (SD) *	3.86 (0.70)	3.81 (0.78)
	Mean score of 5 items: social competence (SD) *	3.99 (0.74)	3.92 (0.81)
	Mean score of 5 items: social resources (SD) *	4.39 (0.56)	4.34 (0.63)
	Mean score of 4 items for: structured style (SD) *	3.41 (0.78)	3.48 (0.83)

Note: Valid cases are 6161 in 2007 and 3576 in 2015 after listwise deletion. * indicates a difference between 2007 and 2015 significant at 0.05 level. ^#^ The same results were reported in Table 1 in [34]. READ–Resilience Scale for Adolescents.

**Table 2 children-09-00948-t002:** Total access to support systems among youth in Norway: 2007 and 2015.

	2007 (N = 7033)	2015 (N = 4531)
Never been in contact with support systems *	81.7%	79.3%
Have ever been in contact with any one or all support systems *	18.3%	20.7%
Have been in contact with only one of the three support systems	13.8%	13.7%
Have been in contact with two support systems *	3.5%	4.9%
Have been in contact with all three support systems *	0.9%	2.1%

Note: * Difference between 2007 and 2015 significant at 0.05 level.

**Table 3 children-09-00948-t003:** Logistic regression of background variables, victimization of violence, and self-injury behaviors in relation to young people’s access to three support systems (ExpB odds ratios (95%CI for ExpB)).

Independent Variables	Dependent Variables					
	Child Welfare Services		Mental Health Services for Children and Youth		Pedagogical Psychology Services	
	2007	2015	2007	2015	2007	2015
Female	1.19 [0.92, 1.52]	1.29 [0.97, 1.71]	**1.57** **[1.24, 1.99]**	**1.30** **[1.05, 1.60]**	1.16 [0.97, 1.38]	0.92 [0.74, 1.13]
Both parents have higher education	0.86 [0.67, 1.12]	**0.59** **[0.43, 0.81]**	1.15 [0.92, 1.42]	0.95 [0.77, 1.18]	0.97 [0.82, 1.15]	0.83 [0.66, 1.04]
Parents live together	**0.31** **[0.25, 0.40]**	**0.29** **[0.22, 0.38]**	**0.54** **[0.44, 0.66]**	**0.52** **[0.42, 0.63]**	**0.69** **[0.59, 0.82]**	**0.78** **[0.64, 0.97]**
Family usually has good financial situation	**0.60** **[0.48, 0.76]**	**0.56** **[0.43, 0.73]**	**0.73** **[0.59, 0.91]**	**0.78** **[0.63, 0.97]**	**0.76** **[0.64, 0.90]**	**0.70** **[0.56, 0.88]**
Both parents are immigrants	**1.97** **[1.43, 2.71]**	**1.78** **[1.20, 2.63]**	**0.58** **[0.39, 0.87]**	**0.54** **[0.36, 0.81]**	0.92 [0.70, 1.22]	0.69 [0.47, 1.02]
No victimization (ref.)						
Victimization of a single form of violence	**1.83** **[1.17, 2.86]**	**1.75** **[1.09, 2.81]**	1.01 [0.71, 1.44]	**1.70** **[1.26, 2.30]**	1.21 [0.95, 1.55]	1.17 [0.87, 1.58]
Victimization of two forms of violence	**3.21** **[2.10, 4.91]**	**2.82** **[1.79, 4.43]**	**1.87** **[1.35, 2.58]**	**2.23** **[1.65, 3.01]**	**1.73** **[1.36, 2.21]**	**1.58** **[1.17, 2.12]**
Victimization of three forms of violence	**4.24** **[2.79, 6.46]**	**6.63** **[4.32, 10.19]**	**2.21** **[1.59, 3.07]**	**3.50** **[2.57, 4.75]**	**2.27** **[1.77, 2.91]**	**2.24** **[1.65, 3.05]**
Victimization of four forms of violence	**7.64** **[4.88, 11.95]**	**8.85** **[5.44, 14.39]**	**3.17** **[2.20, 4.56]**	**4.96** **[3.41, 7.22]**	**2.79** **[2.07, 3.74]**	**3.25** **[2.23, 4.75]**
No self-injury (ref.)						
Have done one of the two forms of self-injury	**1.43** **[1.07, 1.90]**	**1.89** **[1.41, 2.55]**	**2.61** **[2.05, 3.33]**	**2.48** **[1.97, 3.13]**	**2.13** **[1.75, 2.91]**	**1.99** **[1.55, 2.56]**
Have done both forms of self-injury	**2.58** **[1.84, 3.61]**	**5.46** **[3.06, 9.76]**	**9.79** **[7.36, 13.03]**	**16.14** **[9.02, 28.89]**	**4.34** **[3.34, 5.63]**	**5.57** **[3.31, 9.38]**
% correctly predicted	94.7	93.1	93.0	88.4	88.6	89.7
% variance explained (Nagelkerke *R*^2^)	19.8	27.3	22.3	20.4	12.4	10.0

Note: Numbers in bold indicate significant effect at 0.05 level. Missing gender information of 45 cases (0.6%) in 2007 and 65 cases (1.4%) in 2015.

**Table 4 children-09-00948-t004:** Variables of background, severity of adversities, resilience and their relationship to young people’s access to all support systems: descriptive and logistic regression analyses.

Access to Any One or All Supporting Systems No = 0, Yes = 1	Descriptive Analysis				ExpB Odds Ratio [95%CI for ExpB]	
	2007		2015		2007	2015
	No	Yes	No	Yes		
	81.7	18.3	79.3	20.7		
Female = 1 (otherwise = 0) %	55.4	70.4	56.1	66.9		
Both parents have higher education = 1 (otherwise = 0) %	37.2	31.1	37.0	28.8		
Parents live together = 1 (otherwise = 0)	70.3	48.2	68.8	46.7	0.51 [0.44, 0.59]	0.51 [0.42, 0.62]
Family usually has good financial situation = 1 (otherwise = 0) %	74.2	55.6	77.8	61.1	0.73 [0.62, 0.86]	0.74 [0.60, 0.91]
Both parents are immigrants = 1 (otherwise = 0) %	8.6	9.90	9.6	8.1		
Polyvictimization = 3–4 types of violence %	17.8	43.5	13.5	39.5	2.16 [1.74, 2.69]	3.28 [2.47, 4.35]
Severe self-injury = two forms of self-harm (otherwise = 0) %	2.2	16.7	0.4	6.1	4.38 [3.28, 5.84]	11.28 [5.52, 23.07]
Mean score of mental health problems (SD)	1.51 (0.49)	1.97 (0.71)	1.58 (0.52)	1.95 (0.65)	1.93 [1.68, 2.21]	1.42 [1.17, 1.71]
Mean score of resilience dimension: family cohesion (SD)	4.13 (0.72)	3.69 (1.00)	4.21 (0.75)	3.76 (0.98)	0.91 [0.84, 0.98]	0.90 [0.81, 0.98]
Mean score of resilience dimension: personal competence (SD)	3.92 (0.66)	3.54 (0.80)	3.89 (0.74)	3.47 (0.88)		
Mean score of resilience dimension: social competence (SD)	4.02 (0.72)	3.85 (0.82)	3.97 (0.78)	3.68 (0.91)		
Mean score of resilience dimension: social resources (SD)	4.42 (0.52)	4.16 (0.69)	4.40 (0.59)	4.13 (0.76)		
Mean score of resilience dimension: structured style	3.44 (0.78)	3.22 (0.81)	3.53 (0.82)	3.22 (0.87)		
R-Square, % variance explained					24.8	22.6

Note: All descriptive differences of the variables between groups of “No” and “Yes” are significant at the 0.05 level. Only significant associations from the logistic regression analysis are reported here. Stepwise method was used.

## Data Availability

Data are available at Voldprogrammet—Forskningsprogram on vold i nære relas-joner (in English: The Domestic Violence Research Program—A research program about violence in close relationships; https://blogg.hioa.no/voldsprogrammet/forskere/ (accessed on 20 April 2022)). Since Norway is a small country, these survey data must be stripped of several background variables before being fully accessible without restriction. Therefore, as per our usual practice, data are safely stored and accessible from the NOVA—OsloMet Institutional Data Access/Ethics Committee, which grants researchers access if they meet the criteria to access confidential data.

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
