# Peer review of "Access to Public Support Systems Related to Severity of Adversities and Resilience among Adolescents"

_children, 2022, doi:10.3390/children9070948_

Round 1

Reviewer 1 Report

The article addresses a relevant topic, related to the search for social responses for adolescents, based on an analysis of the possibility of victimization and self-assessment of resilience indicators. The study offers data on the prevalence of victimization (including multiple victimization) in young people, as well as contributing to the self-reported identification of protective factors, although the central issue focuses more on contact with structures, which in my view can have very broad interpretations. Therefore, it is necessary that the summary closes by mentioning conclusions reached by the authors, highlighting proposals.

Considering that the study proposes to relate the three central contents (social responses, juvenile victimization and resilience), the theoretical framework seems somewhat modest for such broad research topics. Thus, it is suggested that the theoretical foundation could include other studies, in order to find grounds for the discussion of the results, which is considered to be the weakest part.

In terms of methodology, there are no critical aspects to highlight. The study design is well described. The samples are very satisfactory and the sampling technique is adequate. The collection instruments were presented and are well characterized. The procedures complied with the ethical criteria and are reported (however it was not possible to open the link that informed about the studies). The results confirm a similar pattern from one year to the other studied, but the lack of exposition of concrete objectives (general and specific) does not clarify the reader as to the real purpose of this comparison. What are the hypotheses to be tested? Even the absence of a distinct pattern would be debatable, as much prevention work (primary, secondary, tertiary) at distinct levels may have been done. Has there been no impact of this work? How can this result be explained? What moderating or mediating variables might be included in future studies to ascertain these trends? It is thus considered that the discussion should be enhanced with further reflections and these should be substantiated with more studies and references, which in this part hardly appear.

The results corroborate some results that the literature has already identified (cf. 313-317) and this article could have a better contribution if it problematized some issues. It is expected that those who experience more victimization may have more contact with these structures, but how can we interpret the contact of young people with these services? What kinds of young people? If they are from families whose parents have a higher social and economic level, it is quite possible that contact with public services is reduced in detriment of proven services if necessary? On the other hand, could some young people's use of services have a positive meaning (relative to better access and effectiveness of social responses) or must it be something less positive? Sometimes awareness-raising actions among this population contribute to the demystification of certain services, to the appreciation of their importance, and to the increase in demand.

In short, the work is considered very interesting, and the effective contribution of this article may be in the discussion and conclusions (which should be based on the results obtained) in comparison with the results of other studies.

Reviewer 2 Report

The authors have presented an interesting investigation about the access to three forms of "support" for children experiencing adversity in Norway. The theoretical premise of this article is that accessing these forms of support is an indicator of resilience. It is an interesting standpoint from which to base such an article, and may be particularly relevant to the Norwegian context. However, I think for an international journal, it would be helpful to reflect on the pathway by which people are able to access the services described. For example, is child welfare a self referral process? Or does referral occur after a violent act (as happens in many other countries)? Do any of the services considered have any statutory powers? How is this likely to influence contact with such services? At an international level, some of the services listed might be considered punitive rather than supportive - it would be helpful to understand how they are considered supportive in Norway and how access to them is considered a resilient act (especially if there is a lack of consent in the referral process).

I have some difficulties with the presentation of results. For example, table 3 does not present confidence intervals for the odds ratios. Given the small number of young people who had done both forms of self injury (and the relatively high odds ratio attributed to this variable), I think it is important that the reader is able to understand the width of the confidence intervals involved. I also do not understand why the authors have chosen to only present standardised regression coefficients for table 4, rather than considering the production of odds ratios and confidence intervals.

In lines 381-382 the authors indicate that it is not arbitrary who obtains access to support systems, referencing higher odds ratios for those who had experiences severe adversities as evidence of this. Yet an odds ratio will not provide evidence of the arbitrary nature of service access, it only informs you of the strength of the association between one variable and another. To understand if all of those who experience severe adversity have access to the support system required requires a different type of study from that presented. Lines 384-388 focus on the barriers to accessing support systems, suggesting that these have been lowered and access is available. This statement has not been referenced and requires strong referencing - there is substantial evidence of the impact of high thresholds creating a barrier for young people accessing mental health services.

Rather than taking the approach that this is a theoretical exercise, I think it would be helpful for the investigators to reflect on the reality of what contact with these services means for the young people involved. At present, the paper is situated firmly within a "social resilience" framing, with a view that there is no social cost to the families or young people by being involved with the services. However, the results could be interpreted to suggest that this is not the case - as with many other countries, service delivery appears to be focused on those with fewer resources. While this could be interpreted as positive in a resource constrained environment (those with the most need are being targeted), it could also be interpreted as those with the most resources available work hard to avoid their children from coming in contact with such services (offer other forms of support where available).

Round 2

Reviewer 1 Report

No changes were made to the abstract regarding having a brief concluding sentence. This suggestion should be implemented and is not.

The intro has been improved.

The discussion was worked on, but some of the suggested aspects could be discussed. It is up to the authors to decide how they read the results, even though it may seem like a unidirectional reading. Some of the issues raised could help to have a more critical and reflective discussion with proposals for the future.

The conclusion now looks more like a discussion with debate supported by the literature, when what is expected in the conclusion is a summary of the authors' study findings with recommendations for future studies. At this point, the discussion and conclusion should be reviewed again.

Author Response

Thank you for you kind review and good advices which we have followed in our revision in Track-changes. 

  1. We have added concluding lines to Abstract.
  2. We have moved some texts between Discussion and Conclusion and added some more texts in both Discussion and Conclusion. 

Reviewer 2 Report

The authors are to be congratulated on their diligent approach to addressing the concerns raised.

Author Response

Thank you very much for you kind review and we are grateful for your approval of our previous revision following your good comments and advises!